# A Bayesian Survival Analysis on Long COVID and Non-Long COVID Patients: A Cohort Study Using National COVID Cohort Collaborative (N3C) Data

**DOI:** 10.3390/bioengineering12050496

**Published:** 2025-05-07

**Authors:** Sihang Jiang, Johanna Loomba, Andrea Zhou, Suchetha Sharma, Saurav Sengupta, Jiebei Liu, Donald Brown

**Affiliations:** 1School of Engineering and Applied Science, University of Virginia, Charlottesville, VA 22903, USA; mcu2xn@virginia.edu; 2Integrated Translational Health Research Institute of Virginia (iTHRIV), University of Virginia, Charlottesville, VA 22903, USA; jjl4d@virginia.edu (J.L.); agz5de@virginia.edu (A.Z.); 3School of Data Science, University of Virginia, Charlottesville, VA 22903, USA; ss4jg@virginia.edu (S.S.); ss4yd@virginia.edu (S.S.); deb@virginia.edu (D.B.)

**Keywords:** Bayesian survival analysis, log-normal model, Markov chain Monte Carlo, long COVID, N3C

## Abstract

Since the outbreak of the COVID-19 pandemic in 2020, numerous studies have focused on the long-term effects of COVID infection. On 1 October 2021, the Centers for Disease Control (CDC) implemented a new code in the International Classification of Diseases, Tenth Revision, Clinical Modification (ICD-10-CM) for reporting ‘Post COVID-19 condition, unspecified (U09.9)’. This change indicated that the CDC recognized Long COVID as a real illness with associated chronic conditions. The National COVID Cohort Collaborative (N3C) provides researchers with abundant electronic health record (EHR) data by harmonizing EHR data across more than 80 different clinical organizations in the United States. This paper describes the creation of a COVID-positive N3C cohort balanced by the presence or absence of Long COVID (U09.9) and evaluates whether or not documented Long COVID (U09.9) is associated with decreased survival length.

## 1. Introduction

The outbreak of the COVID-19 pandemic since 2020 has impacted global populations. It is now important to focus on the long-term effects of COVID-19. According to the Centers for Disease Control and Prevention (CDC), Long COVID is broadly defined as signs, symptoms, and conditions that continue or develop after acute COVID-19 infection [1]. Some conditions can last weeks, months, or years. The diagnosis code (U09.9) for Long COVID implemented by the CDC in October of 2021 [2] makes it much easier to identify patients who clinicians believe are suffering from this illness.

With the stewardship of the National Center for Advancing Translational Sciences (NCATS) and data contributions from more than 80 institutions, the National COVID Cohort Collaborative (N3C) [3] is one of the largest publicly accessible collections of clinical data related to COVID-19 patients in the United States, including more than 8 million COVID-positive patients and more than 30 billion rows of electronic health record (EHR) data for cohort studies. Since the implementation of the Long COVID diagnosis code (U09.9), N3C researchers have focused on risk factors and features of Long COVID using machine learning methods [4,5,6,7,8]. However, there is limited research and knowledge related to the mortality risk of Long COVID within the confirmed COVID-19 population. This paper focuses on the survival trends of Long COVID (U09.9) patients along with a matched cohort of COVID-positive control patients and analyzes factors influencing the survival lengths under a Bayesian framework for parameter estimation, and it contributes to understanding Long COVID (U09.9) by quantifying survival trends with probability methods. This study addresses a critical gap in the current Long COVID research by providing empirical estimates of the probability distribution of survival durations among patients diagnosed with Long COVID (U09.9) and without Long COVID in the N3C, and it offers a foundational statistical framework for understanding the characteristics of Long COVID, especially the mortality risk.

## 2. Related Work

We previously developed Bayesian approaches to our survival analysis and also build on prior work related to risk factors for both mortality and Long COVID.

Bayesian survival analysis is a statistical approach focusing on time-to-event data in various fields such as public health and epidemiology, incorporating prior information and uncertainty under Bayesian risk analysis framework [9]. Common methods in Bayesian survival analysis include parametric and semi-parametric models, proportional and non-proportional hazards models, frailty models, cure rate models, etc. Bayesian methods offer flexibility in choosing model structures and allow for uncertainty quantification conveniently, and Markov chain Monte Carlo (MCMC) algorithms provide great tools for sampling from the posterior and predictive distributions.

Various studies on the survival of COVID patients have been conducted since the pandemic. A study in Brazilian patients with COVID showed that old age and cardiovascular disease are associated with higher mortality [10], and a study in patients with COVID admitted to the intensive care unit showed heterogeneity in survival [11]. Another study on COVID and kidney diseases mentioned that acute kidney disease and chronic kidney disease are associated with a higher risk of death, and COVID might lead to chronic kidney disease in survivors [12]. The Charlson Comorbidity Index (CCI) [13] is a weighted sum of common chronic conditions (including kidney and heart disease) that are predictive of inhospital and post-discharge mortality. In our analysis, we use the CCI score as part of our matching criteria to account for the impact of chronic comorbidities on survival.

According to the CDC [1], Long COVID is a disease that can result in chronic conditions that include respiratory and heart symptoms, neurological symptoms, digestive symptoms, etc. For some people, these symptoms can last for weeks, months, or even years. A 2022 study applied a variety of modeling approaches to identify risk factors for Long COVID [4]. Their models consistently showed that middle age (40 to 59 years) and hospitalization at the time of COVID-19 are associated with a higher likelihood of Long COVID diagnosis. In our survival analysis of Long COVID patients and matched controls, these risk factors of Long COVID are used both in the matching process while creating the cohort (e.g., age) and as features in the model (visit severity at the time of COVID).

In contrast to studies on risk factors of contracting Long COVID [4], this paper applies a Bayesian survival analysis approach to individuals diagnosed with Long COVID and other COVID patients, and it identifies the features associated with mortality risks in the Long COVID population. In addition, this paper investigates the distribution of survival lengths of COVID-positive Long COVID patients and COVID-positive control patients; and thus, it provides more comprehensive insights into the survival patterns.

## 3. Methodology

### 3.1. Dataset

We used patient records from N3C that span 1 January 2018 to 14 November 2024 (v186 release). N3C sites’ data were included in the analysis if and only if the site contributed data in 2024. Because data contribution dates still varied between sites, we used matching to control for differences in observation periods between sites.

### 3.2. Cohort Identification and Feature Development

In this study, the cohort consists of a group of COVID patients diagnosed with Long COVID (U09.9) and a matched cohort of COVID patients without this diagnosis. Patients of both groups have either a COVID-19 diagnosis (U07.1), at least one positive result on a SARS-CoV-2 polymerase chain reaction (PCR) or antigen (AG) test, or both. The earliest date of either is used as their COVID index date. All features used in this analysis are derived from the source data using the N3C Logic Liaison Templates [14]. The two-step matching approach [15] described below creates a balanced cohort by assigning each COVID-positive Long COVID patient to a COVID-positive control. In addition to carefully constructing our matched cohort, we also took steps to minimize attrition and immortal time biases as described below.

#### 3.2.1. COVID-Positive Long COVID Group Inclusion Criteria

COVID-19-positive patient as defined with a positive SARS-CoV-2 PCR/AG test or a recorded U07.1 positive diagnosis. The earliest date of either is their COVID index date.Patient was older than or 18 years old at the time of their COVID index.Patient had a U09.9 code in their available health records. The first date of U09.9 was after their COVID index and after 1 October 2021 (date U09.9 was published for use).

#### 3.2.2. COVID-Positive Control Group Inclusion Criteria (No Documented Long COVID)

COVID-19-positive patient as defined with a positive SARS-CoV-2 PCR/AG test or a recorded U07.1 positive diagnosis. The earliest date of either is their COVID index date.Patient was older than or equal to 18 years old at the time of their COVID index.Patient had no U09.9 code in their available health records.Patient was selected as a matched control based on the matching process described below. This ensured a balance of cases and controls with regard to age, COVID era, patient engagement, and healthcare system (site).

#### 3.2.3. Matching Process

In cohort studies, the sample is often not representative of the population at large and, thus, can introduce selection bias [16]. We know this to be true in observational health data where individuals pursuing healthcare are by definition sicker than the population at large. Similarly, patients who more frequently engage with the healthcare institutions are by default more likely to have a death recorded in the electronic health record. In addition, immortal time bias [17] should be taken into consideration in survival analysis. Some of the research participants are ‘immortal’ during a certain time period in that they must survive long enough to receive the intervention being studied. To obtain treated and control groups with similar covariate distributions, choosing well-matched samples of the original treated and control groups is a common and powerful technique to reduce bias due to the covariates. Various matching methods [18] have been applied while using observational data, and below we describe how the two-step matching process generates a balanced cohort while reducing selection bias and immortal time bias.

The first step is a 1-to-5 matching process without replacement using matching variables including patients’ age, health system site id, COVID index date, number of clinical visits prior to COVID, and Charlson comorbidity index (CCI) score [13], resulting in 1 Long COVID patient being matched with 5 COVID positive controls, and satisfying the following conditions:The case and control are from the same health system, minimizing selection bias related to health system population and care practices.The age difference between the case and control is less than or equal to 10 years, minimizing mortality and Long COVID diagnosis biases related to age.The difference in the COVID index date between the case and control is less than or equal to 45 days, minimizing the bias related to the different phases of the pandemic. This also helps ensure similar post-COVID potential observation periods in our survival analysis.The difference in the log of the pre-index number of visits between the case and control must be less than or equal to 1. This ensures a similar level of engagement with the health system, which can both be a proxy for acuity as well as predictive of the capture of mortality events.The difference in the log of CCI score must be less than or equal to 0.5 where the CCI score is based on comorbidities recorded before and up through the COVID index date. This helps balance pre-existing mortality risk between the cases and controls.

The second step is to select 1 final COVID positive control patient out of the 5 matched COVID-positive control patients for each Long COVID patient. In each matching group, we select the control candidate with a documented visit that is closest to the Long COVID patient’s diagnosis date. The visit difference must be no more than 30 days. Therefore, we obtain a 1-to-1 cohort in Long COVID patients and matched COVID-positive control patients with similar attributes and observed survival through Long COVID patient diagnosis.

#### 3.2.4. Survival Timeline Considerations

Although the matching process helps mitigate bias in multiple ways related to patient selection, we also had to be very careful when selecting start and end dates for survival timelines.

According to the CDC, Long COVID starts being identified after several weeks of COVID infection [1]. In this analysis of Long COVID patients and COVID-positive control patients, if the COVID index date was chosen as day 0 of survival analysis, then the fact that Long COVID patients have to survive long enough to receive a U09.9 code would introduce immortal time bias. Thus, we use the U09.9 code date of the Long COVID case as day 0 of the survival timeline for both the cases and their matched control. We drop a pair of patients if anyone in this pair has a death date earlier than the U09.9 code date.

We also recognize that the capture of death data in any electronic health record data is incomplete. In order to enhance our capture of this outcome, we leveraged augmented death data that are made available in N3C through Privacy Preserving Record Linkage (PPRL) [19]. External death dates found in obituaries and government records are linked to N3C patient records using these privacy-preserving techniques. Thus, our N3C death data are a combination of death records as recorded by sites in the electronic health record and harmonized to OMOP [20] and PPRL death records [19]. When more than one death date is reported, the earliest reasonable date is used. Death dates must be within our study period, which ended 14 November 2024 (date of the data extraction used).

The outcome variable, survival length, is defined as the time difference between each patient’s day 0 of survival (U09.9 diagnosis date used for each matched pair) and the patient’s death date or the end of the study period.

### 3.3. Survival Analysis

Survival analyses mainly focus on time-to-event data. Following the notation in [9], we introduce several important variables and functions in survival analysis under Bayesian paradigm. Let *T* be a continuous non-negative random variable representing the survival times of individuals in a population, defined over the interval [0,∞). Let f(t) denote the probability density function (pdf) of *T*, and the cumulative distribution function (cdf) of *T* is(1)$$F\left(t\right)=P\left(T<t\right)=\int_{0}^{t}f\left(u\right)du$$
And the survivor function to describe the probability of surviving till time *t* is(2)S(t)=1−F(t)=P(T>t).
The hazard function h(t), which is the instant rate of failure at time *t*, is defined as(3)$$h\left(t\right)=\lim_{\Delta t\to 0}\frac{P(t<T\leq t+\Delta t|T>t)}{\Delta t}=\frac{f(t)}{S(t)}.$$

Censoring is common in survival data, and it occurs when incomplete information is available about the survival time of some individuals [21]. An observation is said to be right-censored at *c* if the exact value of the observation is not known but only that it is greater than or equal to *c*; an observation is said to be left-censored at *c* if it is known only that the observation is less than or equal to *c*; an observation is said to be interval-censored if it is known only that the observation is in the interval $\left(c_{1},c_{2}\right)$. Type-I censoring and Type-II censoring [22,23] are commonly used in different parametric models as well as in survival analysis.

### 3.4. Bayesian Parametric Models

Survival analysis [9] offers a wide range of parametric distributions, such as exponential, Weibull, and log-normal distributions, each suited to modeling different hazard rate patterns and survival time characteristics. The log-normal distribution is widely used in survival analysis [24,25] because its hazard function is inherently non-monotonic, rising to a peak before declining. Unlike simpler models such as the exponential distribution, which assumes a constant hazard rate, or the Weibull distribution, which assumes a strictly monotonic (increasing or decreasing) hazard, the log-normal model accommodates more complex risk dynamics over time. This flexibility is particularly important given the limited existing research on the survival characteristics of patients with Long COVID (U09.9).

Under Bayesian paradigm [9], given unknown parameters, we first set up a prior distribution and combine the likelihood function with the prior distribution to obtain the posterior distribution of parameters. Markov chain Monte Carlo (MCMC) methods are widely used in sampling from a complicated distribution, such as Gibbs sampling, Metropolis–Hasting algorithm, and Hamiltonian Monte Carlo [26,27]. PyMC [28] is a Python module allowing users to implement Bayesian statistical models with different parameters, prior distributions, and likelihood functions, as well as calculate the numerical results of the posterior estimation of parameters. In this work, PyMC 5.3.0 is applied in a Python 3.8 environment.

Suppose we have independent and identically distributed (i.i.d.) data for survival time $y={\left(y_{1},...,y_{n}\right)}^{T}$ with a right-censor indicator $\nu ={\left(\nu_{1},...,\nu_{n}\right)}^{T}$ where $\nu_{i}=1$ if $y_{i}$ is an observed failure time, and $\nu_{i}=0$ if $y_{i}$ is right-censored. Let D=(n,y,ν); we consider the following models.

#### Log-Normal Model

The log-normal model is a two-parameter model. The survival time $y_{i}$ has a log-normal distribution defined on (0,+∞) with density function, mean, and variance(4)$$\begin{array}{cc} f(y_{i}|\mu ,\sigma ) & ={\left(2\pi \right)}^{-\frac{1}{2}}{\left(y_{i}\sigma \right)}^{-1}\exp\left(-\frac{1}{2\sigma^{2}}{\left(\log y_{i}-\mu \right)}^{2}\right)\\ E(y_{i}) & =\exp(\mu +\frac{\sigma^{2}}{2})\\ \mathrm{Var}(y_{i}) & =\left[\exp\left(\sigma^{2}\right)-1\right]\exp\left(2\mu +\sigma^{2}\right) \end{array}$$
and survival function(5)$$S\left(y_{i}|\mu ,\lambda \right)=1-\Phi \left(\frac{\log y_{i}-\mu }{\sigma }\right)$$
with Φ(.) representing the cdf of the standard normal distribution. The likelihood is(6)$$\begin{array}{cc} L(\mu ,\sigma |D) & =\prod_{i=1}^{n}f{\left(y_{i}|\mu ,\sigma \right)}^{\nu_{i}}S{\left(y_{i}|\mu ,\sigma \right)}^{1-\nu_{i}}\\ \; & ={\left(2\pi \sigma^{2}\right)}^{-\frac{d}{2}}\exp\left(-\frac{1}{2\sigma^{2}}\sum_{i=1}^{n}\nu_{i}{\left(\log y_{i}-\mu \right)}^{2}\right)\\ \; & \times \prod_{i=1}^{n}y_{i}^{-\nu_{i}}{\left(1-\Phi \left(\frac{\log y_{i}-\mu }{\sigma }\right)\right)}^{1-\nu_{i}} \end{array}$$
Let $\tau =1/\sigma^{2}$, and a common prior distribution p(μ,τ) assumes a normal distribution on μ and a gamma distribution on τ. The posterior distribution is given by(7)p(μ,τ|D)∝p(μ,τ)L(μ,τ|D).
To build a regression model, we introduce covariates through μ and write $\mu_{i}=x_{i}^{T}\beta$. Common prior distributions of β include a uniform improper prior and normal prior.

## 4. Results

### 4.1. Cohort Summary

Using v186 release tables (released on 14 November 2024) in N3C and sites with data contribution in 2024, there are 7,223,164 confirmed positive COVID patients, and among them, there are only 74,873 patients with the U09.9 diagnosis code. Cohort characteristics before the matching process and after the matching process are both included below. Table 1 and Table 2 are the attrition tables before the matching process, and Table 3 and Table 4 show characteristics of patients before the matching process and the final cohort.

After the matching process, the final cohort has 94,874 patients in total, with 47,437 Long COVID (U09.9) patients and 47,437 COVID-positive control patients.

### 4.2. Modeling

Following the notations in the methodology section, $y_{i}$ is the survival time of a patient in the cohort, with the distribution(8)$$\begin{array}{cc} y_{i} & ∼\mathrm{LogNormal}(\mu ,\sigma^{2})\\ \tau & =\frac{1}{\sigma^{2}} \end{array}$$
where(9)$$\mu =\beta_{0}+\beta_{1}x_{i1}+\beta_{2}x_{i2}+\beta_{3}x_{i3}$$
and $x_{i1}$, $x_{i2}$, $x_{i3}$ represent three features in the model: whether the patient has Long COVID (U09.9), whether the patient has obesity (BMI greater than or equal to 30 [29]), and whether the patient has mild severity at the time of COVID (mild COVID is defined as no emergency department visits nor hospitalization around COVID index date). The prior distributions of the parameters are as follows:(10)$$\begin{array}{cc} \beta_{0} & ∼N(0,2)\\ \beta_{1} & ∼N(0,2)\\ \beta_{2} & ∼N(0,2)\\ \beta_{3} & ∼N(0,2)\\ \tau & ∼\mathrm{Gamma}(1,0.5) \end{array}$$

### 4.3. Parameter Estimation

Given prior distributions and likelihood specified by the modeling, the posterior of parameters was numerically estimated using Markov Chain Monte Carlo (MCMC) in PyMC [28]. Table 5 shows the posterior mean, standard deviation, and 95% high-density interval of parameters, and Figure 1 shows the density plots and trace plots of parameters in the MCMC sampling process. In the trace plots, all chains follow a similar pattern and occupy a similar region of the parameter space, and the lines look like fuzzy noise with no long flat areas or trends. The density plots are smooth and unimodal with a clear shape of the posterior, and each chain contributes similarly to the overall distribution. As a result, Figure 1 shows that these samples present adequate mixing in the posterior estimation.

## 5. Discussion

According to the posterior estimation of parameters, patients with the Long COVID (U09.9) indicator are more likely to have shorter survival lengths at a 95% significance level, patients with mild visit severity at the time of COVID are more likely to have longer survival lengths at a 95% significance level, and the obesity indicator is not significant at a 95% significance level concerning the survival length.

To diagnose MCMC convergence [30], several statistical and visual tools are commonly used. The Gelman–Rubin diagnostic compares variance within and between chains, with values close to 1.0 indicating convergence. The effective sample size (ESS) assesses the number of effectively independent draws, with a higher ESS suggesting better mixing. Other available tools include spectral density-based methods, the Raftery–Lewis diagnostic, kernel density-based methods, and autocorrelation plots. Figure 1 shows visual evidence for adequate mixing in the posterior estimation, and more quantitative MCMC diagnostic tools could be explored in PyMC.

The Kaplan–Meier estimator is a non-parametric method to estimate the survival probability from lifetime data [31]. Figure 2 shows the Kaplan–Meier curves of two groups in the cohort, and according to the curves, the survival probability of patients in both groups decreases as time goes by, and the Long COVID group has an even lower survival probability than the COVID-positive control group from day 0 of survival till the maximum observed survival length. The significance level of difference between the two groups in survival curves could be further examined using the log-rank test [32].

Since the CDC issued the U09.9 code in October 2021, the maximum possible survival length of a Long COVID patient is less than 1000 days. According to Figure 3, the distribution of survival length is left-skewed. This paper uses a parametric method [9], assuming the observations of survival length are from a log-normal distribution. Other distributions should be taken into consideration to accommodate the left skewness and the non-monotonicity of the hazard function.

In a parametric model, the cumulative distribution function of the survival length is assumed to be differentiable, but this assumption might not hold. Semi-parametric models mainly focus on the baseline hazard or cumulative hazard, and a common example is the piecewise constant hazard model, where the hazard rate might differ in time periods and different subgroups [33,34].

As mentioned in the cohort identification section, the purpose of the matching process and the choice of day 0 of each matched pair in the survival analysis is to create a balanced cohort and control the selection bias and the immortal bias [16,17]. In this analysis, the matching process guarantees that each matched pair of the Long COVID patient and the COVID-positive control patient has similar health risks represented by the CCI score, age, etc. However, we acknowledge other limitations related to the nature of observational health data. For example, because health records are incomplete and some data are not captured in our structured data, some comorbidities are not reflected in our CCI scores. More importantly, the rates of the U09.9 code in the pre-matched sample confirm another work [5] that suggests that Long COVID is underrepresented by this diagnosis code. Computable phenotypes of Long COVID provide another means of identifying Long COVID, making it possible to create a cohort of much more Long COVID patients than U09.9 patients. We elected to avoid using an N3C computable phenotype [35,36] due to the survival bias it would introduce, related primarily to the fact that the post-COVID visit frequency is an important feature in their model.

In a survival analysis, a competing risk, i.e., an event whose occurrence precludes the occurrence of the primary event of interest, could lead to inaccurate results. In studies on cardiovascular disease and depression, respectively [37,38], failure to account correctly for competing events can result in unexpected consequences, including an overestimation of the probability of the event and a mis-estimation of the magnitude of relative effects of covariates on the outcome. In this paper, the reasons for death in Long COVID patients and COVID-positive control patients are largely unavailable, and this remains a limitation in observational survival analysis.

## 6. Conclusions

This survival analysis established a balanced cohort of COVID-positive Long COVID patients and matched COVID-positive control patients with similar comorbidity risks and explored the distribution of the survival length and possible factors influencing the survival length. Our analyses revealed that the Long COVID (U09.9) diagnosis is associated with a shorter survival length at a 95% significance level; mild visit severity at the time of COVID is associated with a longer survival length at a 95% significance level. The obesity indicator is not significant, suggesting that the comorbidity burden was effectively balanced between cases and controls. These results provide a clearer picture of the survival trends of Long COVID (U09.9) patients and control patients, and they help with the further understanding of the clinical implications of Long COVID.

New methods to control selection and immortal bias in survival analyses, new Long COVID computable phenotypes, and other Bayesian parametric and semi-parametric methods in survival analyses can provide opportunities for further investigation.

## Figures and Tables

**Figure 1 bioengineering-12-00496-f001:**
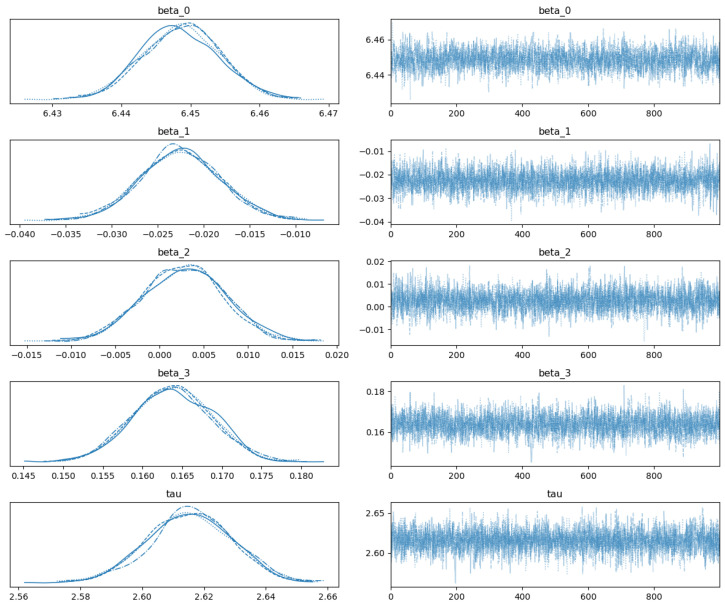
Posterior density plots and trace plots of parameters by MCMC.

**Figure 2 bioengineering-12-00496-f002:**
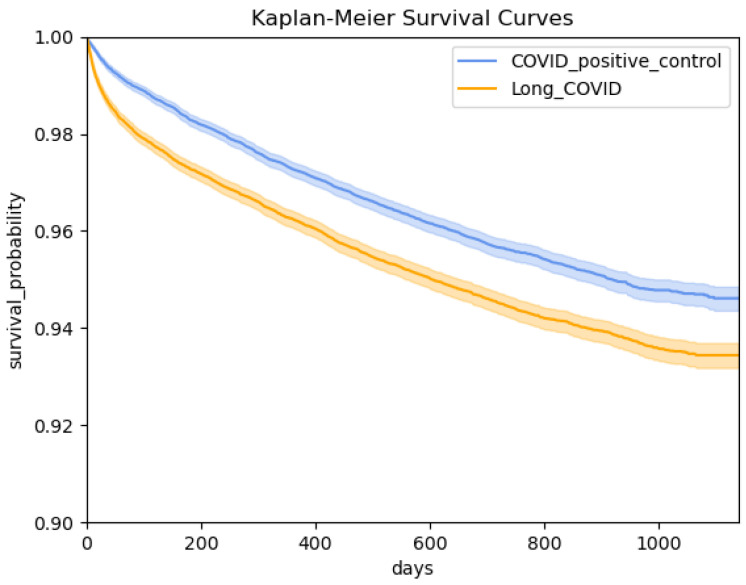
Kaplan–Meier curves of two groups in the cohort.

**Figure 3 bioengineering-12-00496-f003:**
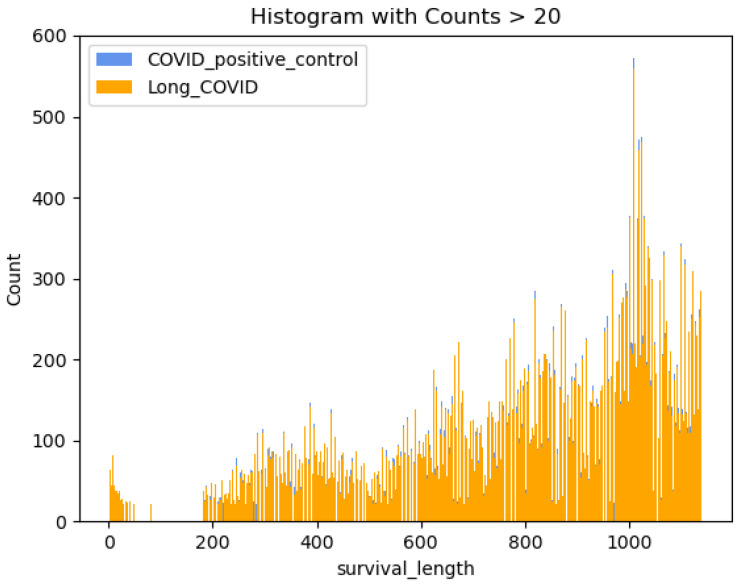
Survival length of the cohort.

**Table 1 bioengineering-12-00496-t001:** Long COVID group attrition table before matching.

Inclusion Criteria	Number of Patients
COVID-positive patients	7,223,164
patients with U09.9 code	74,873
U09.9 date later than COVID index date and 1 October 2021	72,881
age greater than or equal to 18	70,569

**Table 2 bioengineering-12-00496-t002:** COVID-positive control group attrition table before matching.

Inclusion Criteria	Number of Patients
COVID-positive patients	7,223,164
age greater than or equal to 18	6,074,612
patients with no U09.9 code	5,998,558
patients from clinical sites reporting U09.9	5,310,527

**Table 3 bioengineering-12-00496-t003:** Summary table before matching.

	COVID-Positive Control Group	Long COVID Group
number of patients	5,310,527	70,569
age (mean (SD))	48.22 (18.96)	54.87 (16.52)
sex (%)		
Female	3,016,285 (56.8)	46,238 (65.5)
Male	2,288,951 (43.1)	24,297 (34.4)
race and ethnicity (%)		
Asian Non-Hispanic	146,001 (2.7)	1917 (2.7)
Black or African American Non-Hispanic	617,374 (11.6)	7798 (11.1)
White Non-Hispanic	3,409,546 (64.2)	49,671 (70.4)
Other Non-Hispanic	200,365 (3.8)	1390 (2.0)
Hispanic or Latino Any Race	638,650 (12.0)	7061 (10.0)
Missing/Unknown	298,591 (5.6)	2732 (3.9)
CCI category (%)		
0	3,517,745 (66.2)	29,447 (41.7)
1–2	1,040,193 (19.6)	22,266 (31.6)
3–4	372,760 (7.0)	9745 (13.8)
4+	311,516 (5.9)	8849 (12.5)
age category (%)		
18–20	258,611 (4.9)	823 (1.2)
21–45	2,265,771 (42.7)	20,442 (29.0)
46–65	1,679,984 (31.6)	29,612 (42.0)
66+	1,106,161 (20.8)	19,692 (27.9)

**Table 4 bioengineering-12-00496-t004:** Final cohort summary table.

	COVID-Positive Control Group	Long COVID Group
number of patients	47,437	47,437
age (mean (SD))	56.35 (16.22)	56.24 (16.40)
sex (%)		
Female	29,079 (61.3)	31,340 (66.1)
Male	18,344 (38.7)	16,078 (33.9)
race and ethnicity (%)		
Asian Non-Hispanic	1472 (3.1)	1256 (2.6)
Black or African American Non-Hispanic	5671 (12.0)	5254 (11.1)
White Non-Hispanic	32,995 (69.6)	33,928 (71.5)
Other Non-Hispanic	887 (1.9)	904 (1.9)
Hispanic or Latino Any Race	4653 (9.8)	4396 (9.3)
Missing/Unknown	1759 (3.7)	1699 (3.6)
CCI category (%)		
0	18,222 (38.4)	17,006 (35.8)
1–2	14,035 (29.6)	15,447 (32.6)
3–4	8103 (17.1)	7833 (16.5)
4+	7004 (14.8)	7052 (14.9)
age category (%)		
18–20	380 (0.8)	434 (0.9)
21–45	12,134 (25.6)	12,371 (26.1)
46–65	20,006 (42.2)	20,039 (42.2)
66+	14,917 (31.4)	14,593 (30.8)

**Table 5 bioengineering-12-00496-t005:** Posterior estimation of parameters.

Parameter	Mean	Standard Deviation	95% High-Density Interval
$\beta_{0}$ (intercept)	6.449	0.006	[6.439, 6.459]
$\beta_{1}$ (U09.9)	−0.023	0.004	[−0.031, −0.014]
$\beta_{2}$ (obesity)	0.003	0.005	[−0.006, 0.011]
$\beta_{3}$ (mild visit severity)	0.164	0.005	[0.155, 0.173]
τ (precision)	2.616	0.014	[2.588, 2.639]

## Data Availability

Patient data are stored in N3C enclave, DUR-7937888. https://covid.cd2h.org/enclave/ (accessed on 9 April 2025).

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
