# Peer review of "A Bayesian Survival Analysis on Long COVID and Non-Long COVID Patients: A Cohort Study Using National COVID Cohort Collaborative (N3C) Data"

_bioengineering, 2025, doi:10.3390/bioengineering12050496_

Round 1

Reviewer 1 Report

Comments and Suggestions for Authors

The authors use the Bayesian method to determine the survival times of COVID-19 patients. The examination of this disease, which still has a worldwide effect, in terms of survival times, adds value to the article. Using Bayesian analysis is an appropriate approach in estimating uncertain situations. The data set used is detailed, and the inclusion and exclusion criteria for the data are presented. I have a few suggestions to add value to the article.
1. There are many distribution models. The Log-normal Model is preferred in the article. The rationale for using the Log-normal Model should be detailed. Does the survival time only fit the log-normal distribution?
2. What does the difference in the changes given in Figure 2 mean? Briefly explain in terms of survival probability.
3. Summarize the contributions to the literature in the Introduction section.
4. Some references are websites. Indicate the websites. Indicate the access dates.
5. Make sure that all parameters or functions are defined.

Author Response

Comments 1: There are many distribution models. The Log-normal Model is preferred in the article. The rationale for using the Log-normal Model should be detailed. Does the survival time only fit the log-normal distribution?

Response 1: Thanks for pointing this out. Common distributions in survival analysis include exponential, Weibull and log-normal distributions, and exponential distribution has a constant hazard rate, Weibull distribution has a monotonic increasing or decreasing hazard rate, but log-normal distribution has a non-monotonic hazard rate increasing first then decreasing. There is limited literature on the hazard rate of Long COVID (U09.9) patients, and we believe that log-normal describes the hazard rate of Long COVID (U09.9) better than exponential distribution and Weibull distribution. A paragraph and two references were added in Section 3.4. 

Comments 2: What does the difference in the changes given in Figure 2 mean? Briefly explain in terms of survival probability.

Response 2: The explanation is included in the second paragraph in Section 5. The survival probability of patients in both Long COVID (U09.9) group and the COVID positive control group decreases as time goes by, and the Long COVID group has an even lower survival probability than the COVID positive control group from day 0 of survival till the maximum observed survival length. This figure shows that in this cohort, Long COVID patients have a higher mortality risk than COVID positive control patients.

Comments 3: Summarize the contributions to the literature in the Introduction section.

Response 3: A brief summary is added in the last paragraph of Section 1.

Comments 4: Some references are websites. Indicate the websites. Indicate the access dates.

Response 4: Websites in the reference list have been updated with websites and the date of access.

Comments 5: Make sure that all parameters or functions are defined.

Response 5: All the parameters and functions are defined in section 3.4 and section 4.2. 

Reviewer 2 Report

Comments and Suggestions for Authors

This manuscript describes the creation of a COVID positive N3C cohort balanced by the presence or absence of Long COVID (U09.9), through matching of subjects; and applies Bayesian data analysis to evaluate the degree of association documented Long COVID (U09.9) is associated with survival time.
I have several suggestions to improve the presentation of the ideas and results of this manuscript.

Main Comments

  1. The manuscript throughout mentions the use of 5:1 matching of subjects. Please elaborate a bit further on what type of matching method you used (e.g., among the many different matching methods reviewed by Stuart, 2010, Statistical Science). In particular, did you use exact 5:1 matching of subjects? Please clarify.
  2. Following the previous point, it is known that the exact matching of subjects leads to removing subjects from the dataset before data analysis. Therefore, it may be of interest to see how much the results of the analysis differs using optimal full matching of subjects (or alternatively, coarsened exact matching; Iacus, King, Porro 2012), as such a matching method does not remove any subjects from the data set prior to data analysis. Just a suggestion. Though, keeping all the millions of subjects for data analysis would drastically increase the computational complexity of the MCMC algorithm.
  3. Please provide more details on the MCMC algorithm, including the full conditional posterior distributions that are sampled in each sampling iteration of the algorithm.
  4. Line 223:  Here mention how many MCMC samples were generated. Figure 1 suggests that only 1,000 MCMC samples were simulated. Consider generating 10,000 MCMC samples to ensure better MCMC convergence to the posterior distribution of model parameters.
  5. Please provide a link to the relevant software code, so that readers can reproduce the equations and results presented in the manuscript.
  6. The iThenticate report of this submitted manuscript indicates a 54% match of this paper with previous published articles, which is noticeably high. Therefore, consider revising the manuscript to substantially reduced this percentage.

Author Response

Comments 1: The manuscript throughout mentions the use of 5:1 matching of subjects. Please elaborate a bit further on what type of matching method you used (e.g., among the many different matching methods reviewed by Stuart, 2010, Statistical Science). In particular, did you use exact 5:1 matching of subjects? Please clarify.

Response 1: According to Section 3.2.3, we are using an exact match on health system ID (data_partner_id in N3C), and nearest match on CCI, age, COVID index date and number of visits per month. The nearest match here has calipers, meaning the difference of case and control in these variables can't exceed a number, as described in Section 3.2.3.

Comments 2: Following the previous point, it is known that the exact matching of subjects leads to removing subjects from the dataset before data analysis. Therefore, it may be of interest to see how much the results of the analysis differs using optimal full matching of subjects (or alternatively, coarsened exact matching; Iacus, King, Porro 2012), as such a matching method does not remove any subjects from the data set prior to data analysis. Just a suggestion. Though, keeping all the millions of subjects for data analysis would drastically increase the computational complexity of the MCMC algorithm.

Response 2: Among all COVID positive patients in N3C (more than 7 million), only about 1 to 2 percent are Long COVID (U09.9) patients. The exact match on the health system site can't guarantee that each Long COVID patient has available matched controls. 

Comments 3: Please provide more details on the MCMC algorithm, including the full conditional posterior distributions that are sampled in each sampling iteration of the algorithm.

Response 3: The MCMC algorithm in PyMC is No-U-Turn sampler, as showed in reference [27]. No-U-Turn Sampler is a specialized form of the Metropolis-Hastings algorithm, designed as an extension of Hamiltonian Monte Carlo. The No-U-Turn Sampler uses gradient-informed proposals to move in the parameter space, and because of using gradient, there is no fixed sampling distribution in each iteration. 

Comments 4: Line 223:  Here mention how many MCMC samples were generated. Figure 1 suggests that only 1,000 MCMC samples were simulated. Consider generating 10,000 MCMC samples to ensure better MCMC convergence to the posterior distribution of model parameters.

Response 4: We use the default No-U-Turn Sampler in PyMC, with 4 chains, each chain has 1000 tune-in samples and another 1000 samples. The results only keep the 1000 valid samples in each chain. According to the numerical result, the posterior distribution of parameters has only one peak, and in interval estimation we get significance.  https://www.pymc.io/projects/docs/en/stable/api/generated/pymc.sample.html 

Comments 5: Please provide a link to the relevant software code, so that readers can reproduce the equations and results presented in the manuscript.

Response 5: The N3C Data Enclave is a secure platform through which harmonized clinical data provided by contributing members are stored. Anything downloaded outside this platform needs to comply with N3C policies. Researchers can register for the Enclave, and after logging in, all the data and codes are in this project: [RP-1144A9]

https://covid.cd2h.org/enclave/

Comments 6: The iThenticate report of this submitted manuscript indicates a 54% match of this paper with previous published articles, which is noticeably high. Therefore, consider revising the manuscript to substantially reduced this percentage.

Response 6: A previous version of this work has been posted on medRXiv (used data up to June 2024). This version uses more updated data (up to November 2024). 

MedRxiv version: https://doi.org/10.1101/2024.06.25.24309478

Round 2

Reviewer 2 Report

Comments and Suggestions for Authors

The authors adequately addressed nearly all my previous comments. 

The response to Comment 4 could have used more rigorous methods for establishing convergence of the 1,000 MCMC samples. (Or more samples, if 1,000 samples are not enough to establish convergence of the MCMC samples to the posterior distribution). 
The paper could add a sentence mentioning that the trace plot of the 1,000 MCMC samples in Figure 1 shows that these samples present adequate “mixing” (i.e., adequate exploration of the marginal posterior distribution, for each model parameter). This provides one way (perspective) for evaluating MCMC convergence. 

Still, it seems that the authors can take the MCMC convergence analysis to a more rigorous level, by reporting the results of other MCMC convergence diagnostics, such as the Gelman- Rubin statistic among others. Also, the ‘coda’ R package provides various MCMC convergence diagnostics and associated articles and books on the topic. 

Author Response

Comments 1: 

The response to Comment 4 could have used more rigorous methods for establishing convergence of the 1,000 MCMC samples. (Or more samples, if 1,000 samples are not enough to establish convergence of the MCMC samples to the posterior distribution). 
The paper could add a sentence mentioning that the trace plot of the 1,000 MCMC samples in Figure 1 shows that these samples present adequate “mixing” (i.e., adequate exploration of the marginal posterior distribution, for each model parameter). This provides one way (perspective) for evaluating MCMC convergence. 

Still, it seems that the authors can take the MCMC convergence analysis to a more rigorous level, by reporting the results of other MCMC convergence diagnostics, such as the Gelman-Rubin statistic among others. Also, the ‘coda’ R package provides various MCMC convergence diagnostics and associated articles and books on the topic. 

Response 1: Thanks for your comments on MCMC diagnostics. Several sentences are added in section 4.3 to support that Figure 1 shows adequate mixing for the MCMC. The second paragraph in Section 5 and reference [30] are added to address other diagnostic tools for MCMC convergence.

Also, N3C enclave supports both R and Python programming. We preferred PyMC because of our familiarity with Python.